# Activation-aware Probe-Query: Effective Key-Value Retrieval for Long-Context LLMs Inference

## Abstract

Recent advances in large language models (LLMs) have showcased exceptional performance in long-context tasks, while facing significant inference efficiency challenges with limited GPU memory. Existing solutions first proposed the sliding-window approach to accumulate a set of historical **key-value** (KV) pairs for reuse, then further improvements selectively retain its subsets at each step. However, due to the sparse attention distribution across a long context, it is hard to identify and recall relevant KV pairs, as the attention is distracted by massive candidate pairs. Additionally, we found it promising to select representative tokens as probe-Query in each sliding window to accurately represent the entire context, an approach that has been overlooked in the pursuit of effective KV cache eviction. Thus, we propose **ActQKV**, a training-free, **Act**ivation-aware approach that dynamically determines probe-**Q**uery and leverages it to retrieve the relevant **KV** pairs for inference. Specifically, ActQKV monitors a token-level indicator, Activation Bias, within each context window, enabling the proper construction of probe-Query for retrieval at pre-filling stage. To accurately recall the relevant KV pairs and minimize the irrelevant ones, we design a dynamic KV cut-off mechanism guided by information density across layers at the decoding stage. Experiments on the Long-Bench and $\infty$ Benchmarks demonstrate its state-of-the-art performance with competitive inference quality and resource efficiency. Our source code is available at https://anonymous.4open.science/r/ActQKV-DDE1.

## 1 Introduction

With the emergence of large language models (LLMs) capable of handling extended context lengths (Wang et al., 2024b; Achiam et al., 2023; Dubey et al., 2024), researchers are leveraging their advanced information understanding and filtering abilities to tackle various downstream tasks, including web-based search chatbot (Semnani et al., 2023) and document-level question answering (QA) (Lewis et al., 2020). Inevitably, the context length has increased significantly, even surpassing the models' context limitations. However, the computational complexity of attention mechanism (Vaswani, 2017) grows quadratically $O(N^2)$ with the context length $N$ during inference. Specifically, each token from context will be embedded into Query (Q) and interactive with Key (K) and Value (V) embedded from all the $N$ tokens using attention weights, making the whole time and memory complexity $O(N^2)$ for the process. Even worse, during inference, new tokens are generated one by one while each generation triggers a $O(N^2)$ computation, leading to an $O(N^2 + MN^2)$ to generate an output of length $M$. Therefore, efficiency is a critical challenge in the deployment of long-context LLMs (Li et al., 2024a).

To handle this issue, the sliding window mechanism has been proposed to segment the input sequence into content blocks and incrementally convert them into a key-value (KV) cache for reuse (Beltagy et al., 2020). During inference, the model computes the KV vectors only for the current window and integrates them with the existing KV cache, thereby reducing redundant KV computations, leading to an $O(N^2 + MN)$ complexity. Building on this mechanism, recent works (Xiao et al., 2024a; Li et al., 2024b; Hao et al., 2025; Fountas et al., 2025) focus on retrieving top-$k$ relevant KV pairs in conjunction with current tokens for preserving long-term contextual dependencies, where further reduces the complexity to $O(kN + kM)$. In this process, the queries from current window are

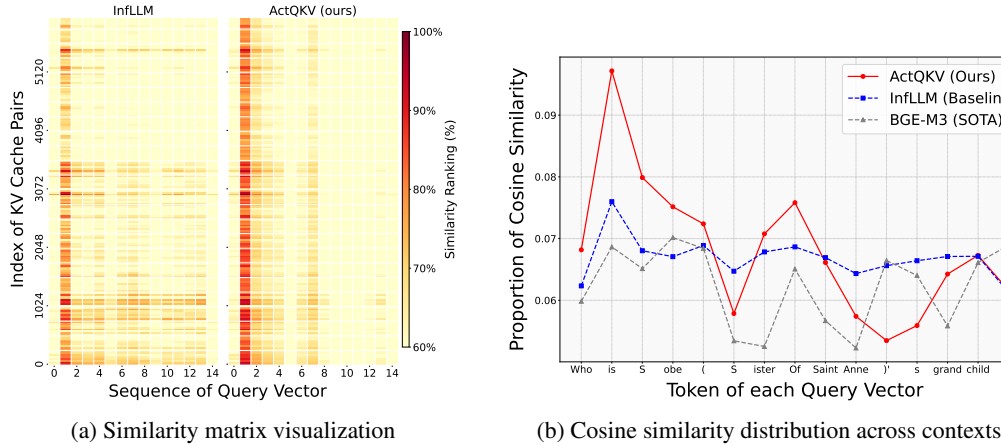

(a) Similarity matrix visualization  (b) Cosine similarity distribution across contexts

Figure 1: Visualization of query vector status within probe-Query compared between ActQKV and InfLLM: **(a)** Token-level similarity matrix, **(b)** Cosine similarity distribution across contexts. We simply display the states of a question "Who is Sobe (Sister of Saint Anne)'s Grandchild?" and the context from a window of size 256 in the last transformer layer. Our probe-Query shows closer alignment with BGE-M3 (Chen et al., 2024) embeddings, while InfLLM exhibits uniform similarity distribution neglecting anchor prioritization. We further conduct a quantitative verification in Appendix D to analyze the effectiveness of the probe-Query in recovering the anchor distribution.

typically compressed as a **probe-Query** for relevant KV retrieval. However, this probe-Query setting often fails to highlight those anchor tokens with critical activation signals, which are rare and essential to represent long context within the sliding window.

To address this challenge, we first investigate the similarity relationship between the composition of the probe-Query and KV cache. Under sparse attention patterns (see Fig. 1a), the query vectors generated by InfLLM (the left) are disordered. In this scenario, each query vector influences the semantics of probe-Query, which makes the combined representation nondescript. To clearly demonstrate this nondescript (see Fig. 1b), the blue line employs a widely used mean pooling technology along KV dimension to represent the probe-Query. It is evident that the probe-Query fails to capture the distinctions because attention is distracted by all tokens instead of focusing on the anchors. Therefore, such a nondescript probe-Query is hard to represent semantic of question and unsuitable for effective KV retrieval.

Motivated by these observations, we argue that only a subset of anchor tokens within the context window plays a dominant role in representing probe-Query for retrieval. In this paper, we propose ActQKV, a training-free method that incorporates sliding window attention, which mainly involves two stages: matching and recall of relevant KV pairs. **In KV matching stage**, we construct the probe-Query for each context window to retrieve the relevant KV pairs in a streaming manner. To effectively estimate the anchor tokens during inference, we employ a window-level activation-aware strategy to monitor the fluctuation of query values for each token. Recognizing that the scarce outlier features is a critical factor affecting model performance (Wang et al., 2024a; Wu et al., 2024), we designate activated query vectors with prominent activation bias to dominate the representation of probe-Query for accurate retrieval, as shown in red line of Fig. 1b. **In KV recall stage**, due to the irregular distribution of KV pairs across layers, a fixed threshold often fails to yield optimal retrieval results. In particular, the decoding stage, which is highly sensitive to factual correctness, can be adversely affected by irrelevant KV pairs, potentially leading to hallucinations and degrading the overall quality of the generated text. Therefore, we introduce a KV cut-off mechanism that dynamically adjusts the number of selected pairs based on information density of each layer. Under a constrained KV budget, this mechanism enhances the recall of relevant KV pairs while reduces the introduction of irrelevant ones.

Our contributions are summarized as follows:

- Motivated by attention distraction phenomenon, we introduce an activation-aware probe-Query that efficiently emphasizes anchor tokens essential for accurately matching KV pairs. It is the first exploration to extract long-context representations for KV retrieval within the query vector.

- To further eliminate irrelevant KV pairs and recall the relevant, we design a dynamic KV cut-off mechanism guided by information density across layers during the decoding stage. This method effectively enhances the model's factual filtering ability for reasoning QA.

- Our ActQKV outperforms existing SOTA KV retrieval-based methods with just 2K KV budget on two benchmarks, achieving up to a 16× KV reduction and 10.4% accuracy improvement compared to using the full cache setting with a 2K budget on LongBench.

## 2 RELATED WORKS

**KV cache retrieval** (Adnan et al., 2024; Zhang et al., 2023; Xiao et al., 2025) has become a critical optimization strategy aimed at reducing memory usage, minimizing inference latency and improving overall throughput in long-context LLMs inference.

Recent studies employ a sliding window mechanism to address challenges in long-text inference, where tokens outside the window are stored in the cache and only used when needed for the current window. To accelerate the retrieval of essential KV, several approaches have proposed index-based methods that organize and access the KV cache at the block or cluster level, enabling efficient querying and extraction. InfLLM (Xiao et al., 2024a) maintains the full KV cache in blocks and uses a hierarchical storage strategy to facilitate long-sequence processing. This framework employs CPU-GPU memory orchestration, keeping essential KV and computational units in GPU memory while offloading less frequently accessed units to CPU memory. Q-LLM (Li et al., 2024b) enhances long-sequence processing by prioritizing memory related to task descriptions. This approach mimics human reading behavior: first reading the question, then searching for the answer in the context.

In contrast to methods which use uniform KV block sizes, TokenSelect(Hao et al., 2025) is based on the observation of sparsity in non-continuous attention patterns. It uses the Query-Key dot product to assess the importance of each KV cache stored at the token level. For each query, they dynamically calculates the importance of past KV caches per head at the token level and selects the most important tokens through a soft voting mechanism across heads. EM-LLM (Fountas et al., 2025) dynamically segments incoming tokens into episodic events, employing a hybrid retrieval mechanism that combines semantic similarity matching with temporal context to efficiently access relevant KV cache segments. Additionally, some researchers focus on KV cache budget allocation across layers (Cai et al., 2024; Yang et al., 2024) and heads (Feng et al., 2024; Fu et al., 2025) due to the hierarchical architecture of LLMs.

Most methods overlook the importance of probes for retrieval, especially given the fact that LLMs are not optimized for retrieval tasks. Therefore, this realization inspires our further exploration of probe-Query construction in this paper.

## 3 BACKGROUND

In this section, we introduce the two-stage inference of long-context LLMs using sliding window attention (in Sec. 3.1), and then define the problem of KV Retrieval (in Sec. 3.2).

### 3.1 SLIDING WINDOW ATTENTION WITH KV CACHE

Given an input sequence $\mathbf{X}$, the generation of the output sequence $\mathbf{Y}$ during LLMs inference can be divided into two stages: pre-filling the input $\mathbf{X}$ and decoding the output $\mathbf{Y}$.

To handle long sequences input of tasks, existing works (Xiao et al., 2024b;a; Li et al., 2024b) use sliding window attention to process the text iteratively. In this mechanism, the lengthy input sequence $\mathbf{X}$ is partitioned into $T$ windows, denoted as $\mathbf{W} = \{\mathbf{w}^1, \ldots, \mathbf{w}^T\}$, $\mathbf{W} \in \mathbb{R}^{T \times m}$ and $m$ indicates the window size (see Fig. 2(a)). To reduce computational costs, the model processes each window sequentially and stores the historical key-value pairs in a cache (i.e., $\mathbf{K}_{\text{cache}}$ and $\mathbf{V}_{\text{cache}}$) for future reuse (see Fig. 2(b)).

**During $t$-th pre-filling step** ($t \leq T$), the model utilizes the KV cache $\mathbf{K}_{\text{cache}}^{t-1}$ and $\mathbf{V}_{\text{cache}}^{t-1}$ from the historical sequence $\mathbf{W}[: t-1]$ to compute the attention output $\mathbf{O}^t \in \mathbb{R}^{m \times d}$ for the current $m$ window tokens $\mathbf{w}^t \in \mathbb{R}^m$ as follows:

$$\mathbf{O}^t = \text{Attention}\left(\mathbf{Q}^t, \left[\mathbf{K}^t, \mathbf{K}_{\text{cache}}^{t-1}\right], \left[\mathbf{V}^t, \mathbf{V}_{\text{cache}}^{t-1}\right]\right), \tag{1}$$

where the triplet $\mathbf{Q}^t = \{\mathbf{q}_i^t\}_{i=1}^m, \mathbf{K}^t = \{\mathbf{k}_i^t\}_{i=1}^m, \mathbf{V}^t = \{\mathbf{v}_i^t\}_{i=1}^m \in \mathbb{R}^{m \times d}$ represents the generated attention vectors, each corresponds to $m$ tokens with $d$ hidden dimensions. To further save GPU memory, current methods select partial KV cache $\mathbf{K}^*$ and $\mathbf{V}^*$ for inference, denoted as:

$$\mathbf{O}^t = \text{Attention}\left(\mathbf{Q}^t, \left[\mathbf{K}^t, \mathbf{K}^*\right], \left[\mathbf{V}^t, \mathbf{V}^*\right]\right), \tag{2}$$

where $\mathbf{K}^* \subseteq \mathbf{K}_{\text{cache}}^{t-1}$ and $\mathbf{V}^* \subseteq \mathbf{V}_{\text{cache}}^{t-1}$.

**During $t$-th decoding step** ($t > T$), the model generates the output sequence $\mathbf{Y}$ token-by-token. Unlike pre-filling, the model uses only one single query vector $\mathbf{q}^t \in \mathbb{R}^{1 \times d}$ along with corresponding key and value vectors $\mathbf{k}^t, \mathbf{v}^t \in \mathbb{R}^{1 \times d}$ to predict one next token $y^t \in \mathbf{Y}$ in each step. Its corresponding attention output $\mathbf{o}^t \in \mathbb{R}^{1 \times d}$ can be computed as:

$$\mathbf{o}^t = \text{Attention}\left(\mathbf{q}^t, \left[\mathbf{k}^t, \mathbf{K}^*\right], \left[\mathbf{v}^t, \mathbf{V}^*\right]\right). \tag{3}$$

**After the $t$-th step**, the newly generated key-value pairs will be stored in the cache (see Fig. 2(e)), updating it as demonstrated below:

$$\mathbf{K}_{\text{cache}}^t, \mathbf{V}_{\text{cache}}^t = \mathbf{K}_{\text{cache}}^{t-1} \cup \mathbf{K}^t, \mathbf{V}_{\text{cache}}^{t-1} \cup \mathbf{V}^t, \tag{4}$$

where $\cup$ denotes the concatenation operation and the tensors of cache can be saved in either CPU or GPU memory. In general, saving in the CPU can significantly reduce the memory usage of the GPU. Note that $\mathbf{K}^t = \mathbf{k}^t$ and $\mathbf{V}^t = \mathbf{v}^t$ are $1 \times d$ dimensions during decoding.

## 3.2 PROBLEM SETTING

During long-context inference in LLMs, the historical key-value pairs are essential for maintaining long-range dependencies and overcoming window size limitations. Given a cache comprising $\mathbf{K}_{\text{cache}}^{t-1}$ and $\mathbf{V}_{\text{cache}}^{t-1}$, the objective of KV retrieval is to identify the top-$k$ relevant subset $\mathbf{K}^*$ and $\mathbf{V}^*$ using the probe-Query $\mathbf{Q}_{\text{probe}}^t$ for the $t$-th inference step (Xiao et al., 2024a; Fountas et al., 2025; Hao et al., 2025), as described below:

$$\mathbf{K}^*, \mathbf{V}^* = \mathbf{K}_{\text{cache}}^{t-1}[I^*], \mathbf{V}_{\text{cache}}^{t-1}[I^*],$$

$$I^* = \arg\max_{\substack{I \subset [m], \\ |I| = k}} \sum_{i \in I} \left( \frac{\mathbf{Q}_{\text{probe}}^t \cdot \mathbf{K}_{\text{cache}}^{t-1}[i]^\top}{\|\mathbf{Q}_{\text{probe}}^t\| \times \|\mathbf{K}_{\text{cache}}^{t-1}[i]\|} \right), \quad [m] = \{1, 2, \ldots, m\}, \tag{5}$$

where $\mathbf{Q}_{\text{probe}}^t \in \mathbb{R}^{1 \times d}$ denotes the overall representation of window context $\mathbf{w}^t$ and $k$ is the number of selected KV. These two factors significantly impact the factual relevance of the retrieved KV index $I^*$ for each transformer layer inference.

## 4 METHODS

In this section, we first present the overall framework of our ActQKV, as illustrated in Fig. 2. We then demonstrate our two-stage approach: the Activation-aware Probe-Query Construction for KV matching (in Sec. 4.1)and the Dynamic KV Cut-off Mechanism for KV recall (in Sec. 4.2).

## 4.1 ACTIVATION-AWARE PROBE-QUERY

To identify the relevant KV pairs, we leverage the query vectors of each window to construct the attention-aware probe-Query for retrieval. The primary distinction between our activation-aware probe-Query and other representation methods lies in the emphasis on identifying anchor tokens that effectively represent the entire context of the window for KV matching. The main challenge is to accurately distinguish and activate these tokens.

Figure 2: Illustration of our ActQKV. Sliding window attention stores historical KV pairs in a cache and reuses them for subsequent window inference. Based on this, ActQKV first identifies the anchor tokens within the window and then constructs the activation-aware probe-Query. This probe-Query is subsequently used to retrieve the top-k relevant KV pairs from the cache during the pre-filling stage. During the decoding stage, the cut-off mechanism dynamically adjusts the number of recalled KV pairs based on the distribution of key-values at each layer, ensuring the inclusion of relevant pairs while minimizing the influence of irrelevant ones. The cache can be stored in the CPU and transferred to the GPU when needed. All our contributions are highlighted in red.

Formally, given a subset of context $\mathbf{w}^t = \{x_1^t, \ldots, x_m^t\}$ extracted from a long sequence $\mathbf{W}$, we obtain the hidden states $\{z_i^t\}_{i=1}^m = \{f(x_i^t)\}_{i=1}^m$ at each transformer layer, where $m$ denotes the window size and $f$ denotes the function mapping tokens to corresponding states. Intuitively, hidden states that deviate significantly from their statistical mean (i.e., $\bar{\mathbf{z}}^t$) can be considered that they are from anchor tokens compared to others. Specifically, token $x_1^t$ is deemed more essential than $x_2^t$ for the quality of generation, as indicated by previous works (Wang et al., 2024a; Sun et al., 2024; Pang et al., 2024), if:

$$\|\bar{\mathbf{z}}^t - f(x_1^t)\| > \|\bar{\mathbf{z}}^t - f(x_2^t)\|, \tag{6}$$

where $\|\cdot\|$ is distance metrics.

Building on the aforementioned paradigm Eq. 6, we propose an **Activation Bias** to distinguish the importance of each query vector within a window context. For the query vectors of the $t$-th pre-filling window $\mathbf{Q}^t = \{\mathbf{q}_1^t, \ldots, \mathbf{q}_m^t\}$ in each layer, we first compute the token-level bias $\mathbf{\Phi}^t = \{\phi_1^t, \ldots, \phi_m^t\}$, with $\mathbf{\Phi}^t \in \mathbb{R}^{m \times d}$, to estimate the energetic degree within $\mathbf{Q}^t$ as follows:

$$\phi_j^t = \frac{(\mathbf{q}_j^t - \bar{\mathbf{z}}^t)^2}{\sigma^2}, \tag{7}$$

where $\sigma^2$ and $\bar{\mathbf{z}}^t \in \mathbb{R}^{1 \times d}$ represent the variance and mean of the query vectors respectively, computed as follows:

$$\sigma^2 = \frac{\sum_{i=1}^t \sum_{j=1}^m \left(\mathbf{q}_j^i - \bar{\mathbf{z}}^t\right)^2}{mt - 1}, \bar{\mathbf{z}}^t = \frac{\sum_{i=1}^t \sum_{j=1}^m \mathbf{q}_j^i}{mt}. \tag{8}$$

Based on the above estimated degree, we can construct the probe-Query $\mathbf{Q}_{\text{probe}}^t$ for KV matching by reassigning the activated weights of each query vector according to the activation bias $\mathbf{\Phi}^t$:

$$\mathbf{Q}_{\text{probe}}^t = \sum_{j=1}^m \frac{\|\phi_j^t\|_1}{\|\mathbf{\Phi}^t\|_1} \mathbf{q}_j^t. \tag{9}$$

Our object is to enhance the weight of query vectors for those anchor tokens. With this activated probe-Query, we can match more precise KV pairs $\mathbf{K}^*$ and $\mathbf{V}^*$ that contain semantically relevant information for pre-filling stage Eq. 2.

## 4.2 DYNAMIC KV CUT-OFF MECHANISM

During the decoding stage, the quality of the predicted answer greatly depends on the top-$k$ relevant pairs $\mathbf{K}^*$ and $\mathbf{V}^*$. However, due to the sparse and irregular attention pattern across each layer, the selection of $k$ KV pairs is highly sensitive to the probe-Query $\mathbf{Q}_{\text{probe}}^t = \mathbf{q}^t$. Therefore, we propose a KV cut-off mechanism to dynamically determine $k$ based on information density assessment for $L$ transformer layers. Compared to the preset threshold, this mechanism dynamically removes redundant KV pairs and improves the recall of relevant ones within a limited KV budget.

In the $t$-th decoding step, we first calculate the similarity scores $\mathbf{S}^\ell = \{s_1^\ell, \ldots, s_n^\ell\}$ between the probe-Query $\mathbf{Q}_{\text{probe}}^t$ and the cache of key vectors $\mathbf{K}_{\text{cache}}^{t-1}$ for the $\ell$-th transformer layer, where $n = |\mathbf{K}_{\text{cache}}^{t-1}|$. The similarity scores are computed using cosine similarity as follows:

$$s_i^\ell = \frac{\mathbf{Q}_{\text{probe}}^t \cdot \mathbf{K}_{\text{cache}}^{t-1}[i]}{\|\mathbf{Q}_{\text{probe}}^t\| \times \|\mathbf{K}_{\text{cache}}^{t-1}[i]\|}. \tag{10}$$

Then, we apply the softmax function to normalize them and convert them into probabilities.

Based on the similarity distribution $\mathbf{S}^\ell$, we define the information density $\Theta^\ell$ for the $\ell$-th layer using the entropy function as follows:

$$\Theta^\ell = -\sum_{i=1}^n \frac{e^{s_i^\ell}}{\sum_{j=1}^n e^{s_j^\ell}} \log\left(\frac{e^{s_i^\ell}}{\sum_{j=1}^n e^{s_j^\ell}}\right), \tag{11}$$

where a uniform distribution results in a higher information density $\Theta^\ell$ compared to more concentrated distributions.

Now with the information density, we focus on dynamically assigning the budget instead of a fixed value $k$ for each layer. Given a total budget $\mathbf{B}_{kv}$, we process from shallow to deep layers in the order of transformer computation to avoid decoding delays. Consequently, for the $\ell$-th layer in the $t$-th decoding step, the budget $\mathbf{B}^\ell$ can be estimated as follows:

$$\mathbf{B}^\ell = \frac{\Theta^\ell}{\Theta^\ell + \sum_{j=\ell+1}^L \bar{\Theta}^j} \times \mathbf{B}_{kv}, \tag{12}$$

where $\mathbf{B}_{kv}$ is initialized as $L \times k$ and updated by $\mathbf{B}_{kv} \leftarrow \mathbf{B}_{kv} - \mathbf{B}^\ell$ after processing the $\ell$-th layer, and $\bar{\Theta}^j$ denotes the mean $\Theta^\ell$ for the remaining unprocessed layers. In this part, we aim to assign a larger budget to layers with higher information density, where many KV pairs are potentially relevant to the probe-Query $\mathbf{Q}_{\text{probe}}^t$ for the $t$-th decoding step. Conversely, for layers with lower density, the relevant KV pairs with higher similarity are more prominent, making the irrelevant pairs more likely to be discarded. Based on the above Eq. 12, the denominator, which adds $\Theta^\ell$ to the cumulative average density $\sum_{j=\ell+1}^L \bar{\Theta}^j$ of the remaining layers, quantifies the overall contribution of both the current and subsequent layers. A higher ratio indicates that the current layer holds a more significant portion of the relevant KV pairs, justifying a larger allocation. Compared to using a fixed threshold for retrieval, this dynamic KV cut-off mechanism eliminates redundant KV pairs and improves the recall of relevant ones within the limited KV budget.

In summary, we present our two-stage method separately, where the activation-aware probe-Query module guarantees the quality of historical KV pairs and the cut-off mechanism effectively utilizes them. The entire process is depicted in Algorithm 1 as shown in Appendix C.

## 5 EXPERIMENTS

In this section, we first present the experimental setup of this paper (in Sec. 5.1). Then we demonstrate the logical reasoning and factual retrieval ability of our ActQKV in long-context inference through two widely-used benchmark (in Sec. 5.2). Finally, we conduct the ablation study (in Sec. 5.3) and reveal the influence of our method (in Sec. 5.4).

### 5.1 EXPERIMENTAL SETUP

**Datasets and Implementation Details.** We utilize 21 tasks from two widely used long document benchmarks: Long-Bench (Bai et al., 2023) and $\infty$-Bench (Zhang et al., 2024) for evaluation.

| Method
KV Budget | LLaMA3-8B-inst
full context | Infinite
2K | Stream
2K | InfLLM
2K | QLLM
2K | TSLLM
2.5K | EMLLM
4K | ActQKV
2K |
|---|---|---|---|---|---|---|---|---|
| NarrativeQA | 19.85 | 16.47 | 15.12 | 19.41 | 25.60 | 22.44 | 22.50 | **27.04** |
| Qasper | 42.36 | 32.01 | 31.72 | 41.27 | 39.12 | 40.74 | **44.95** | 40.42 |
| MultiFieldQA | 41.03 | 31.63 | 30.99 | 45.89 | 48.30 | 47.73 | 48.79 | **50.70** |
| HotpotQA | 47.38 | 34.73 | 35.26 | 44.97 | 49.91 | 50.33 | 49.19 | **51.37** |
| 2WikiMQA | 39.20 | 29.22 | 30.59 | 36.27 | 39.63 | 31.38 | 38.08 | **42.07** |
| Musique | 22.96 | 13.50 | 13.64 | 19.73 | 25.03 | 24.53 | 25.19 | **33.40** |
| GovReport | 29.94 | 27.84 | 27.83 | 30.68 | 29.80 | **32.56** | 30.85 | 32.00 |
| QMSum | 21.45 | 19.91 | 20.14 | 21.36 | 22.23 | **23.50** | 22.77 | 23.06 |
| MultiNews | 27.51 | 27.36 | 27.37 | 27.87 | 27.85 | **27.92** | 27.28 | 27.26 |
| Trec | 74.00 | - | - | 57.50 | 55.50 | 67.50 | **73.50** | 69.50 |
| TriviaQA | 90.50 | **88.07** | 87.35 | 88.03 | 87.70 | **92.22** | 90.91 | 85.68 |
| SAMSum | 42.30 | 36.93 | 35.97 | 34.86 | 34.97 | 42.16 | **43.24** | 40.10 |
| PassageRetrieval | 62.50 | 23.50 | 23.50 | 85.25 | 88.00 | 87.00 | 86.00 | **94.50** |
| LCC | 60.83 | 60.42 | 58.15 | 58.17 | 58.37 | 58.86 | 60.44 | **62.04** |
| RepoBench-P | 49.14 | **64.95** | 62.97 | 62.01 | 61.04 | 51.24 | 44.88 | 61.92 |
| Average | 44.73 | 36.18 | 35.76 | 43.98 | 46.20 | 46.67 | 47.24 | **49.40** |

Table 1: Long-Bench (avg. 31K tokens) (Bai et al., 2023). The comparison of results based on LLaMA3-8B-inst (AI@Meta, 2024) are conducted from the works (Li et al., 2024b; Hao et al., 2025; Fountas et al., 2025). Our results are highlighted in teal and best results are indicated in bold.

Specifically, Long-Bench has a 95% sequence length of 32K, while ∞-Bench averages about 122K in sequence length. We utilize LLaMA3-8B-inst (AI@Meta, 2024) and Qwen2.5-7B-Instruct (Team, 2024) as our base models with maximum input lengths of 8K and 32K, respectively. In each inference step, we reuse only 2K KV pairs and store the remaining pairs in the Cache Management system, following the settings of InfLLM. This approach consumes approximately 19 GB of VRAM in our experiments. Inspired by previous works, we retain 64 attention sinks and 512 KV pairs from current context, and adapt the task description into probe-Query. Consequently, the budget for retrieved KV $k$ is 1,472. These KV pairs are organized into 46 chunks, with each chunk containing 32 pairs. The sliding window size is set to 256. More details about the datasets and experimental setup is available in Appendix B.

**Baseline Methods**    The objective of ActQKV is to effectively retrieve key-value pairs for long-context inference in LLMs. To achieve this, we evaluate two prominent baseline methods: **(a)** static KV selection and **(b)** KV retrieval. **(a):** Infinite (Lin et al., 2024) employs global and local attention masks to broaden the attention scope, while Stream (Xiao et al., 2024b) ensures efficient inference by retaining attention sinks and KV pairs from recent tokens. **(b):** InfLLM (Xiao et al., 2024b) searches for KV pairs associated with the currently processed tokens, enabling the capture of long-distance dependency relationships. QLLM (Li et al., 2024b) focuses on KV memory relevant to the task description to process long sequences. TokenSelect (TSLLM) (Hao et al., 2025) incorporates the token-level weight of KV cache per-head for KV retrieval. EMLLM (Fountas et al., 2025) integrates key aspects of human episodic memory and event cognition into KV cache. Notably, all the methods described above are **training-free**.

## 5.2 Main Experiment Results

We first utilize Long-Bench to evaluate the long-context reasoning capabilities of ActQKV, and then test the fact retrieval ability using ∞-Bench. We report the results based on Llama-3-8B-Instruct, and the others can be found in Appendix B and Appendix C.

**Long-Bench.**    We present the results in Tab. 1. (1) ActQKV achieves an average score of 49.40, surpassing the full context setting (31K tokens) by 4.67 points while utilizing only 2K tokens. This highlights the **efficiency** of its key-value retrieval method in handling long-context inference with a significantly smaller KV budget. (2) Compared to the static KV selection methods Infinite and Stream, ActQKV excels in capturing critical information required for reasoning tasks. (3) In comparison to SOTA KV retrieval methods such as TSLLM and EMLLM, our activation-aware retrieval approach achieves the best results, with improvements of +5.8% and +4.6%, respectively. Notably, for tasks

like 2WikiMQA and Musique, ActQKV shows substantial gains, demonstrating the effectiveness of activation-aware retrieval in capturing long-term dependencies by recalling fewer KV pairs (e.g., only with 80% and 50% budget).

| Method | KV | ∞-Bench (214K tokens) | | | | | | |
|--------|--------|-------|-------|-------|-------|-------|-------|-------|
| | Budget | C.D. | M.F. | MC | R.KV | R.P | R.N | Avg. |
| InfLLM | 2k | 22.59 | 26.86 | 33.19 | 80.80 | 100.0 | 28.64 | 48.68 |
| QLLM | 2k | 23.10 | 27.37 | 34.50 | **84.00** | 100.0 | 27.63 | 49.43 |
| TSLLM | 2.5k | 27.41 | 28.29 | **45.85** | 40.00 | 100.0 | **97.29** | 56.47 |
| EMLLM | 8k | 31.73 | 17.14 | 40.61 | 5.00 | 100.0 | 99.49 | 49.00 |
| ActQKV | 2k | **42.86** | **29.43** | 38.22 | 46.20 | **100.0** | 93.90 | **58.43** |

Table 2: ∞-Bench (avg. 122K tokens) (Zhang et al., 2024). The results comparison based on LLaMA3-8B-inst. Our results are highlighted in teal and the best are indicated in bold.

∞-**Bench.** Each sample in this benchmark has almost infinite length (avg. 122K), where the key lies in whether factual evidence can be found from the context. As shown in Tab. 2, our ActQKV obtains the best result 58.43 and outperforms the SOTA KV retrieval methods even with a smaller KV budget. Especially compared to the token-level retrieval method TSLLM, our approach sets the minimum retrieval unit as a chunk. Although larger chunks may seem less granular, our probe-Query effectively compensates for this, enhancing 3.5% performance while simultaneously reducing both time and space complexity from $O(N)$ to $O(m)$. This demonstrates that our method can efficiently recall relevant KV pairs even with coarser granularity.

## 5.3 ABLATION STUDIES

| Method | KV | LongBench Categories | | | | | | |
|--------|--------|------|------|------|------|------|------|------|
| | Budget | SQA | MQA | Sum | FSL | Ret | Cod | Avg. |
| InfLLM | 2k | 38.5 | 36.9 | 27.0 | 69.0 | 84.0 | 53.2 | 47.0 |
| TSLLM | 2.5k | 37.0 | 35.4 | 28.3 | **67.3** | 87.0 | 51.2 | 46.7 |
| EMLLM | 8k | 39.3 | 37.7 | 27.0 | **69.2** | 87.5 | 50.3 | 47.2 |
| w/o DCM | 2k | **40.3** | 40.7 | **27.5** | 63.1 | **98.0** | 61.5 | 48.8 |
| w/o APQ | 2k | 39.7 | 42.1 | 27.4 | 64.3 | 94.5 | 61.7 | 49.2 |
| ActQKV | 2k | 39.4 | **42.3** | 27.4 | 65.1 | 94.5 | **62.0** | **49.4** |

Table 3: The ablation study of our method ActQKV, where **A**ctivated **P**robe-**Q**uery (APQ) for KV matching and **D**ynamic **C**ut-off **M**echanism (DCM) for KV recall. We use the mean pooling to represent probe-Query in w/o DCM as same as InfLLM and QLLM.

In this subsection, we present ablation studies shown in Tab. 3 to evaluate two key components of our method: the Activation-aware Probe-Query $Q_{\text{probe}}^t$ (APQ, see Sec. 4.1) and the Dynamic Cut-off Mechanism (DCM, see Sec. 4.2).

When using APQ for key-value (KV) pair matching, our method attains a comparable score of 48.8, especially getting the best result 98.0 in retrieval tasks. These results demonstrate that the APQ component effectively captures the semantic context of the window for KV matching, outperforming conventional mean pooling approaches. Moreover, the incorporation of DCM, which dynamically determines the number of KV pairs to recall at each layer, further enhances the model's ability of irrelevant information filtering. Overall, our approach employs a two-stage KV retrieval process following the traditional information retrieval paradigms: first, an initial retrieval stage identifies potentially relevant KV pairs; subsequently, a refined recall stage optimizes the selection process, achieving a peak performance of 49.4. And the more analysis of model robustness and the effectiveness of activation-aware functions are detailed in the Appendix D.

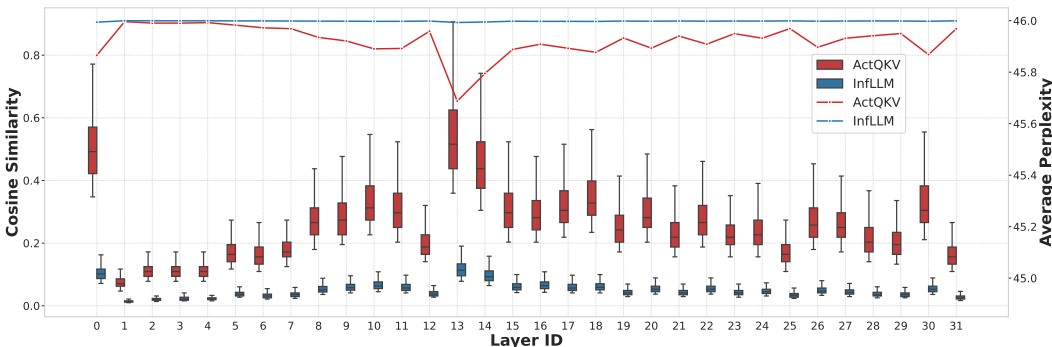

Figure 3: Analysis of the top-$k$ (avg. k=1,472) most relevant KV pairs for each inference step across layers. We randomly select 50 samples from Long-Bench and filter out those with a length less than 8K. In each layer, we calculate 35,180 similarity scores generated by our ActQKV and InfLLM respectively. Each score is calculated based on a probe-Query and a chunk containing 32 KV pairs. The average perplexity is calculated based on the perplexity within the scores of each sample.

## 5.4 ANALYSIS OF RETRIEVED KV PAIRS

In this subsection, we compare the retrieved KV pairs from our ActQKV and InfLLM methods to evaluate the specific impact of our proposed approach. To facilitate this comparison, we present the distribution of cosine similarity scores and average perplexity in Fig. 3 and analyze the following:

**Cosine Similarity.** The box of cosine similarity clearly shows that ActQKV consistently achieves higher similarity scores across most layers compared to InfLLM. This outcome can be attributed to the activation-aware query (probe-Query) we introduced, which more effectively captures the underlying semantic information of the window context for each inference step. Furthermore, the enlargement of the box plots indicates that the distribution of similarities becomes more dispersed. This suggests that our probe-Query covers a broader semantic space, thereby resulting in a more robust KV retrieval process. The greater spread in the similarity values also reflects the model's ability to account for a wider range of relevant KV pairs, ultimately enhancing the precision and adaptability of the retrieval process across different contexts.

**Average Perplexity.** With respect to average perplexity, ActQKV consistently shows lower perplexity scores compared to InfLLM which maintains a value of around 46.0. This indicates that ActQKV yields more coherent and predictable results across the all layers. Notably, in layers 0 and 13, we notice significant differences, with ActQKV showing more variation than InfLLM. This suggests that our retrieval method can flexibly adapt to the characteristics of different layers. By reducing perplexity, ActQKV improves the ability to discriminate relevant KV pairs from irrelevant ones, resulting in more coherent and less uncertain historical information for long-context inferences.

In addition, we provide extended experiments in the Appendix D, including (i) quantitative verification of the probe–oracle alignment, (ii) cross-model comparisons on multiple benchmarks, and (iii) ablations on activation-aware functions and dynamic KV recall. These results consistently validate the robustness and generality of our proposed method across settings.

## 6 CONCLUSION

In this paper, we present ActQKV, a training-free method to KV retrieval efficiency for long-context LLMs inference. The primary challenge in KV retrieval stems from the inherent vagueness of existing probe-Query, which inadequately filter irrelevant KV pairs. To address this limitation, we develop an activation-aware probe-Query construction strategy and a layer-wise KV cut-off mechanism to effectively match and recall the relevant KV pairs. We hope this work can inspire the broader research for LLMs representation methods, leading to improved long-context information filtering capabilities akin to specialized embedding models.

# 7 ETHICS STATEMENT

Throughout the development and execution of this work, we strictly adhered to ethical guidelines established by the broader academic and open-source community. All datasets and models utilized are publicly available. There are no conflicts of interest among the authors involved in this research. Our approach aligns with ethical AI practices, prioritizing trust, accountability, and responsible research.

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

## A   THE COMPLEXITY OF LLMs INFERENCE

| Mechanism | Pre-filling Complexity | Decoding Complexity | Overall Complexity |
|---|:---:|:---:|:---:|
| Standard Attention | $O(N^2)$ | $O(N^2 + MN^2)$ | $O(N^2 + MN^2)$ |
| Sliding Window with KV Cache | $O(N^2)$ | $O(N^2 + MN)$ | $O(N^2 + MN)$ |
| KV Retrieval for Long-Context Inference | $O(kN)$ | $O(kN + kM)$ | $O(kN + kM)$ |

Table 4: Complexity analysis of different methods. Our ActQKV is belong to the KV retrieval method and the complexities are highlighted in teal.

In this section, we focus on the attention computation and analyze the complexity of exiting methods shown in Tab. 4 as follows:

**Standard Attention Mechanism.** Under the standard attention mechanism, during the pre-filling stage, each token in the input sequence undergoes attention calculations with all other tokens, resulting in a time complexity of $O(N^2)$. In the decoding stage, as the context grows, the complexity of generating each new token increases accordingly. When generating the $t$-th token, the length of the context to be processed is $N + t$, so the total time complexity of the decoding stage is $O(\sum_{t=1}^{M} M(N + t)^2)$, which is approximately $O(N^2M + M^3)$. Since the decoding length $M$ is usually much smaller than the input sequence length $N$, the overall complexity can be simplified to $O(N^2 + MN^2)$.

**Sliding Window Mechanism with KV Cache.** The sliding window mechanism divides the input sequence into several windows of fixed size, each with a size of $m$. During the pre-filling stage, the processing complexity of the tokens within each window is $O(m^2)$, and the interaction complexity between the KV caches of the windows is approximately $O(N)$, so the overall time complexity is $O(\frac{N}{m} \times m^2) = O(mN)$, which is equivalent to $O(N^2)$ when the window size m is constant and linearly dependent on N. In the decoding stage, the decoding of each new token only needs to interact with the m tokens in the current window and some tokens in the adjacent windows, resulting in a total time complexity of $O(MN)$. Overall, the time complexity can be simplified to $O(N^2 + MN)$.

**KV Retrieval for Long-Context Inference.** When using the method of Top-k retrieval combined with the sliding window, the pre-filling stage divides the input sequence into windows of fixed size. During the processing of the tokens in each window, only the interaction with the top k most relevant key-value pairs is performed, so the complexity of the pre-filling stage is $O(\frac{N}{m} \times (m + k))$, which can be approximated as $O(kN)$ if the window size m and the retrieval range k meet certain conditions. In the decoding stage, the prediction of each new token only needs to interact with the top-$k$ most relevant key-value pairs, with a time complexity of $O(kM)$. Overall, the time complexity is simplified to $O(kN + kM)$.

## B   DETAILS IN LONG-BENCH AND ∞-BENCH

Long-Bench (95% sequence length is 32K) focuses on tasks that involve reasoning, such as question answering, summarization, few-shot learning, retrieval, and coding. The groups of datasets are categorized as follows: **Single-doc QA**: NarrativeQA, Qasper, MultiFieldQA; **Multi-doc QA**: HotpotQA, 2WikiMQA, Musique; **Summarization**: GovReport, QMSum, MultiNews; **Few-shot Learning**: TREC, TriviaQA, SAMSum; **Retrieval**: PassageRetrieval; **Code**: RepoBench-P. And ∞-Bench (avg. length of 200K) emphasizes factual retrieval, covering domains such as code, mathematics, multiple-choice questions, and general retrieval tasks. The statistics and evaluation metrics of datasets are detailed in Tab. 5 and Tab. 6.

## C   IMPLEMENTATION DETAILS

All experiments were implemented using PyTorch and performed on two NVIDIA A800 80GB GPUs. In all experiments in this paper, we use standard greedy decoding to ensure reliable results.

| Dataset | ID | Source | Avg len | Metric | Language | #data |
|---|---|---|---|---|---|---|
| *Single-Document QA* | | | | | | |
| NarrativeQA | 1-1 | Literature, Film | 18,409 | F1 | English | 200 |
| Qasper | 1-2 | Science | 3,619 | F1 | English | 200 |
| MultiFieldQA-en | 1-3 | Multi-field | 4,559 | F1 | English | 150 |
| *Multi-Document QA* | | | | | | |
| HotpotQA | 2-1 | Wikipedia | 9,151 | F1 | English | 200 |
| 2WikiMultihopQA | 2-2 | Wikipedia | 4,887 | F1 | English | 200 |
| MuSiQue | 2-3 | Wikipedia | 11,214 | F1 | English | 200 |
| *Summarization* | | | | | | |
| GovReport | 3-1 | Government report | 8,734 | Rouge-L | English | 200 |
| QMSum | 3-2 | Meeting | 10,614 | Rouge-L | English | 200 |
| MultiNews | 3-3 | News | 2,113 | Rouge-L | English | 200 |
| *Few-shot Learning* | | | | | | |
| TREC | 4-1 | Web question | 5,177 | Accuracy (CLS) | English | 200 |
| TriviaQA | 4-2 | Wikipedia, Web | 8,209 | F1 | English | 200 |
| SAMSum | 4-3 | Dialogue | 6,258 | Rouge-L | English | 200 |
| *Retrieval* | | | | | | |
| PassageRetrieval-en | 5-1 | Wikipedia | 9,289 | Accuracy (EM) | English | 200 |
| *Code Completion* | | | | | | |
| LCC | 6-1 | Github | 1,235 | Edit Sim | Python/C#/Java | 500 |
| RepoBench-P | 6-2 | Github repository | 4,206 | Edit Sim | Python/Java | 500 |

Table 5: An overview of the dataset statistics in LongBench (Bai et al., 2023). Avg len (average length) is computed using the number of words for the English (code) datasets and the number of characters for the Chinese datasets. Accuracy (CLS) refers to classification accuracy, while Accuracy (EM) refers to exact match accuracy.

| Task | Annotation | # Ex. | Avg Len |
|---|---|---|---|
| Ret.PassKey | Auto | 590 | 122.4K/2 |
| Ret.Number | Auto | 590 | 122.4K/4 |
| Ret.KV | Auto | 500 | 121.1K/22.7 |
| En.MC | Human | 229 | 184.4K/5.3 |
| Code.Debug | Human | 394 | 114.7K/4.8 |
| Math.Find | Auto | 350 | 87.9K/1.3 |

Table 6: Data statistics of $\infty$-Bench (Zhang et al., 2024). The columns indicate whether the annotation was auto-generated or done by humans, the number of examples, and the average length (input/output) in tokens.

Our framework integrates InfLLM's representative token selection mechanism with activation-aware probe queries, where top 4 tokens per KV chunk. We highlight two core innovations:

- **KV Retrieval Enhancement**:

$$\text{similarity- based metrics} \left( \underbrace{\text{Mean-pooling}(Q_{\text{sliding-windows}})}_{\text{InfLLM}}, \text{representative-KV} \right)$$

$$\Rightarrow \quad \text{similarity-based metrics} \left( \underbrace{\text{Activation}(Q_{\text{sliding-windows}})}_{\text{Our ActQKV}}, \text{representative-KV} \right) \tag{13}$$

- **Dynamic Budget Allocation**:

$$\text{selection}(\text{top-}k, \text{similarity metrics})$$
$$\Rightarrow \text{dynamical-selection}(\text{information-density}, \text{top-}k, \text{similarity-metrics}) \tag{14}$$

**Overall Workflow**: The overall pipeline of our ActQKV can be described as follows:

---

**Algorithm 1:** Effective KV Retrieval for Long-context LLMs Inference

---

**Input** : $L$: Total number of transformer layers; $\mathbf{Q}_{\text{probe}}^t$: Probe-Query for the $t$-th step;
  $\mathbf{K}_{\text{cache}}^{t-1}$: Cache of key vectors for the $t-1$-th step; $B_{kv}$: Initial KV budget ($L \times k$)
**Output :** $\mathbf{K}^*$ and $\mathbf{V}^*$: Selected KV pairs for inference
**for** $\ell \leftarrow 1$ **to** $L$ **do**
  **for** $i \leftarrow 1$ **to** $n$ **do**
    $s_i^\ell \leftarrow \frac{\mathbf{Q}_{\text{probe}}^t \cdot \mathbf{K}_{\text{cache}}^{t-1}[i]}{\|\mathbf{Q}_{\text{probe}}^t\| \times \|\mathbf{K}_{\text{cache}}^{t-1}[i]\|}$;
  **end**
  $\mathbf{P}^\ell \leftarrow \text{Softmax}(\mathbf{S}^\ell)$; $\Theta^\ell \leftarrow -\sum_{i=1}^n P(s_i^\ell) \log P(s_i^\ell)$;
**end**
**for** $\ell \leftarrow 1$ **to** $L$ **do**
  $B^\ell \leftarrow \frac{\Theta^\ell}{\Theta^\ell + \sum_{j=\ell+1}^L \bar{\Theta}^j} \times B_{kv}$; $B_{kv} \leftarrow B_{kv} - B^\ell$;
**end**
**for** $\ell \leftarrow 1$ **to** $L$ **do**
  Sort $\mathbf{K}_{\text{cache}}^{t-1}$ based on $\mathbf{P}^\ell$ in descending order; Select top $\mathrm{E}^\ell$ KV pairs; Add selected KV pairs to $\mathbf{K}^*$ and
  $\mathbf{V}^*$;
**end**
**return** $\mathbf{K}^*$, $\mathbf{V}^*$;

---

# D FURTHER ANALYSIS

## D.1 QUANTITATIVE VERIFICATION

Table 7: Spearman correlation between Activated Probe-based Query and oracle block relevance scores.

| Probe Strategy | Median $\rho$ | IQR |
|---|---|---|
| ActQKV (Activation Bias) | **0.73** | 0.68–0.78 |
| Mean Pooling (InfLLM-style) | 0.54 | 0.49–0.59 |

We evaluate whether the activation-bias probe $\mathbf{Q}_{\text{probe}}^t$ can faithfully recover the oracle anchor block distribution across $t = 1, \ldots, 100$ sampled queries. Let $\mathbf{q}_i$ denote the $i$-th token embedding of the $t$-th query and $\mathbf{c}_j$ the embedding of the $j$-th context block, encoded using BGE-M3. The oracle relevance score of block $\mathbf{c}_j$ is defined as

$$s_j^{\text{oracle}} = \max_i \cos(\mathbf{q}_i, \mathbf{c}_j), \tag{15}$$

and sorting $\{\mathbf{c}_j\}$ by $s_j^{\text{oracle}}$ produces the oracle ranking.

For the probe distributions, we consider two variants. (**ActQKV**) The probe vector is constructed using activation bias $\phi_i$:

$$\mathbf{Q}_{\text{probe}}^t = \sum_{i=1}^{|W|} \frac{\|\phi_i\|_1}{\sum_{k=1}^{|W|} \|\phi_k\|_1} \mathbf{h}_i, \quad \mathbf{Q}_{\text{mean}}^t = \frac{1}{|W|} \sum_{i=1}^{|W|} \mathbf{h}_i, \tag{16}$$

where $|W|$ is the number of tokens in the query. The relevance score of each block under a probe is

$$s_j^{\text{probe}} = \cos(\mathbf{Q}^t, \mathbf{c}_j), \tag{17}$$

where $\mathbf{Q}^t$ denotes either $\mathbf{Q}_{\text{probe}}^t$ or $\mathbf{Q}_{\text{mean}}^t$.

To measure the agreement between the oracle and probe distributions, we compute the Spearman correlation directly on the scores:

$$\rho = \frac{\sum_{j=1}^N \left(s_j^{\text{oracle}} - \bar{s}^{\text{oracle}}\right) \left(s_j^{\text{probe}} - \bar{s}^{\text{probe}}\right)}{\sqrt{\sum_{j=1}^N \left(s_j^{\text{oracle}} - \bar{s}^{\text{oracle}}\right)^2 \sum_{j=1}^N \left(s_j^{\text{probe}} - \bar{s}^{\text{probe}}\right)^2}}, \tag{18}$$

where

$$\bar{s}^{\text{oracle}} = \frac{1}{N}\sum_{j=1}^{N} s_j^{\text{oracle}}, \quad \bar{s}^{\text{probe}} = \frac{1}{N}\sum_{j=1}^{N} s_j^{\text{probe}}.$$

Across 100 queries, the ActQKV probe achieves a median correlation of **0.73**, whereas mean pooling attains 0.54, representing a 19% absolute improvement as shown in Tab. 7. This demonstrates that the activation-bias probe more faithfully recovers the oracle anchor block distribution.

## D.2 MODEL COMPARISON

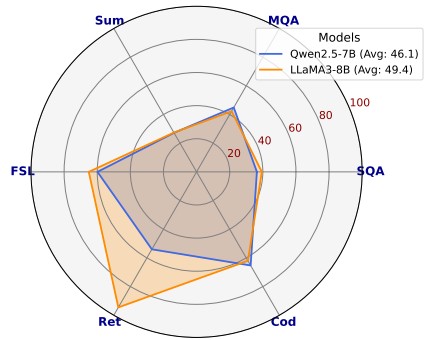 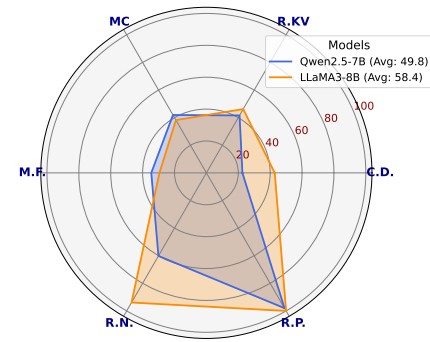

(a) Long-Bench (Bai et al., 2023).    (b) ∞-Bench (Zhang et al., 2024).

Figure 4: Performance comparison based on different models: LLaMA3-8B (AI@Meta, 2024) v.s. Qwen2.5-7B (Team, 2024).

| Method | KV Budget | Model | SQA | MQA | Sum | FSL | Ret | Cod | Avg. |
|--------|-----------|-------|-----|-----|-----|-----|-----|-----|------|
| ActQKV | 2k | Mistral-v0.2-7B | **28.40** | **25.20** | **27.02** | **59.60** | 46.75 | 46.45 | **38.02** |
| Q-LLM | 2k | Mistral-v0.2-7B | 20.14 | 19.97 | 26.32 | 57.00 | **49.67** | **50.21** | 35.06 |
| InfLLM | 2k | Mistral-v0.2-7B | 27.76 | 21.60 | 26.17 | 57.07 | 40.29 | 49.84 | 33.55 |

Table 8: LongBench Evaluation Results on Mistral-v0.2-7B. Comparative evaluation results on Mistral-v0.2-7B with 2K KV cache budget. Our method achieves superior average performance while maintaining competitive results across different task categories.

We conduct experiments on Long-Bench (Bai et al., 2023) and ∞-Bench (Zhang et al., 2024) using LLaMA3-8B (AI@Meta, 2024) and Qwen2.5-7B (Team, 2024), as illustrated in Fig. 4. And also, the experiments using Mistral-v0.2-7B are shown in Tab. 8

For LLaMA3-8B (AI@Meta, 2024), the model achieves state-of-the-art (SOTA) performance across tasks in both Long-Bench and ∞-Bench, demonstrating its versatility, particularly in factual retrieval and code-related tasks. In contrast, although Qwen2.5-7B (Team, 2024) does not match the performance of LLaMA3-8B across all categories, it exhibits substantial improvements over the baseline. The most significant performance drop is observed in the Retrieval tasks, where Qwen2.5-7B underperforms relative to LLaMA3-8B. This highlights a challenge in handling retrieval-related aspects of the benchmarks. Nevertheless, Qwen2.5-7B consistently outperforms the baseline in these tasks, underscoring the effectiveness of our approach, even though it does not yet match the top-performing model in retrieval. However, Qwen2.5-7B excels in code-related tasks, even surpassing LLaMA3-8B in this domain. This demonstrates the model's proficiency in handling complex, domain-specific tasks, such as those encountered in RepoBench-P. While Qwen2.5-7B shows some weaknesses in retrieval, its performance in other specialized areas is either competitive or superior.

Overall, while Qwen2.5-7B shows a decrease in retrieval task performance compared to LLaMA3-8B, it still surpasses the baseline, confirming the efficacy of our method, ActQKV. And ur ActQKV achieves the best experimental results with the Mistral-v0.2-7B model.

### D.3 THE EFFECTIVENESS OF ACTIVATION-AWARE FUNCTIONS

The **Activation Bias** defined in Eq. (6) establishes a theoretical foundation for our activation-aware selection mechanism. To empirically validate its effectiveness, we conduct comprehensive ablation studies to verify the numerical sensitivity of energetic degree $\Phi^t$ in Eq. (9) and information density $\Theta^\ell$ in Eq. (12). Specifically, we try to **reverse** the **impl**ementation of Eq. (6) as follows:

- **Energetic Degree** $\Phi^t \leftarrow \frac{1}{\Phi^t}$ , **and give higher weight for lower degree**:
- **Information Density** $\Theta^\ell \leftarrow \frac{1}{\Theta^\ell}$ , **and give more budget for lower density**:

The results of ablation studies are shown as follows:

| Method | LongBench Accuracy (%) | | | | |
|---|---|---|---|---|---|
| | SQA | MQA | FSL | Cod | Avg. |
| ActQKV | 39.4 | 42.3 | 65.1 | 62.0 | 49.4 |
| w/o APQ&DCM | 38.5 | 36.9 | 64.0 | 59.7 | 47.0 |
| Reverse Impl. | 36.5 | 36.0 | 60.1 | 58.6 | 44.0 |

Table 9: Sensitivity of energetic degree $\Phi^t$ and information density $\Theta^\ell$.

In Tab. 9, we can see that compared with ActQKV and foundation (w/o APQ&DCM), the results with reverse implementation are worse. The significant performance degradation (12.3%) under reverse implementation conditions further demonstrates the effectiveness of our methodological design.

### D.4 DYNAMIC KV PAIRS RECALL

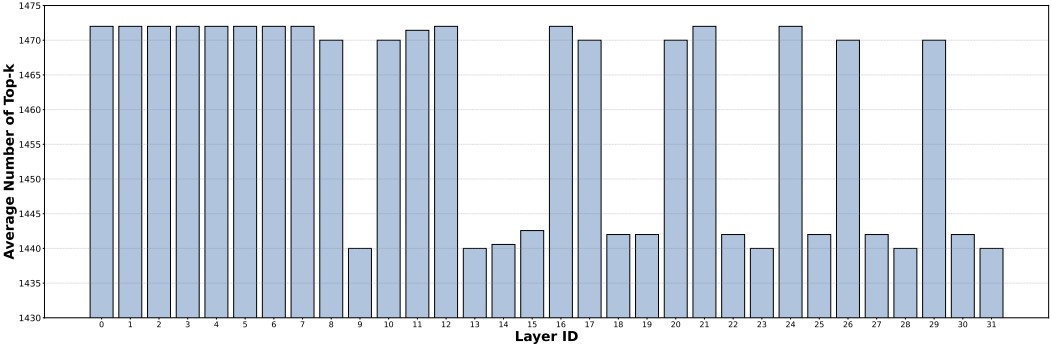

Figure 5: Average number of relevant KV pairs recalled for each layer in decoding stage based on LLaMA3-8B-inst (AI@Meta, 2024). We randomly select 50 samples from Long-Bench and filter out those with a length less than 8K.

Our approach employs a layer-wise key-value cut-off mechanism and an activation-aware probe-Query construction strategy to more effectively match and recall relevant KV pairs. As shown in Fig. 5, we report the average number of relevant KV pairs recalled for each layer.

The results in Fig. 3 and Fig. 5 demonstrate that our method, ActQKV, adapts to the varying distributions across layers, ensuring a robust and efficient retrieval process. Notably, in layer 13, which exhibits the lowest perplexity of similarity scores and receives the smallest KV budget, our method fully aligns with the objectives outlined in Eq. (12). This consistency allows LLMs to effectively process long-context information for long-context inference.

While the per-layer variation appears modest ( 2%), we observe significant cumulative effects when processing long sequences. For instance, in a 25K-token sample divided into 100 sliding windows, the DCM module dynamically adjusts KV recall decisions across layers, ultimately producing meaningfully distinct attention patterns compared to static approaches. This is evidenced by the 2.2% average performance gain of our full method over the w/o DCM baseline in ablation studies.

## D.5   LATENCY PERFORMANCE

| Method | Sequence Length | | Latency (seconds) | | | Speed |
|---|---|---|---|---|---|---|
| | Input (K) | Output | Prefill | Decoder | Total | Relative to Stream |
| Stream | 10 | 71 | 3.15 | 10.46 | 13.74 | 1.00 |
| InfLLM | 10 | 71 | 7.76 | 6.76 | 14.61 | 0.94 |
| ActQKV | 10 | 71 | 8.64 | 18.78 | 27.50 | 0.50 |
| Stream | 100 | 71 | 33.01 | 11.48 | 44.57 | 1.00 |
| InfLLM | 100 | 71 | 61.76 | 8.75 | 70.59 | 0.63 |
| ActQKV | 100 | 71 | 91.05 | 21.62 | 112.76 | 0.40 |
| Stream | 500 | 71 | 162.89 | 10.50 | 173.69 | 1.00 |
| InfLLM | 500 | 71 | 307.76 | 9.83 | 317.67 | 0.55 |
| ActQKV | 500 | 71 | 474.60 | 25.58 | 500.27 | 0.35 |

Table 10: Latency comparison on different input lengths. Latency measurements were conducted on NVIDIA A800 80GB GPUs with FP16 precision based on the `transformers` framework. Stream serves as the latency baseline (Relative=1.0).

Benchmark results indicate that ActQKV consistently exhibits higher latency than InfLLM across all evaluated sequence lengths, with total inference time approximately 1.6–1.9× that of the current SOTA baseline. The additional overhead primarily arises from its core innovation—activation-aware KV cache retrieval—which requires roughly 19GB of VRAM for long-context inputs. Nevertheless, ActQKV delivers substantial gains in long-context reasoning accuracy, demonstrating the effectiveness of our design. In latency-tolerant applications such as medical report analysis, this trade-off between efficiency and performance may be well justified. Future research will investigate hybrid architectures to mitigate inference overhead while preserving accuracy.

## E   USAGE OF LLMS

This manuscript utilized LLMs for manuscript polishing, code formatting, and as a backbone in experiments. The LLM did not contribute to the research design, data analysis, or the substantive content of the work. We use GPT-4o and DeepSeek-R1 to polish our paper.

## F   LIMITATIONS

Our method achieves promising performance to enhance the relevant KV pairs retrieval for long-context LLMs inference. And we believe that the interpretability of the retrieved KV pairs requires further exploration in future works. Unlike non-autoregressive architectures in embedding models, the auto-regressive architecture of LLMs results in the semantics of current tokens being influenced by historical KV pairs. When processing a long context all at once, this interaction makes it difficult to separate the semantics from various events because the retrieved key-value pairs mostly show historical information. This introduces challenges in interpreting the retrieval results. Meanwhile, future work will focus on hybrid architectures to reduce inference time while preserving accuracy.

