# OpenReview forum: "Activation-aware Probe-Query: Effective Key-Value Retrieval for Long-Context LLMs Inference"
_ICLR.cc/2026/Conference — ICLR 2026 Conference Withdrawn Submission_

### Official Review · Reviewer_8isy · 2025-10-23

**Soundness:** 3
**Presentation:** 3
**Contribution:** 3
**Rating:** 4
**Confidence:** 4

**Summary:**

This paper introduces ActQKV, a training-free framework for efficient key-value (KV) retrieval in long-context LLM inference. It proposes an activation-aware probe-query (APQ) that emphasizes highly activated “anchor” tokens to better represent window-level context, and a dynamic KV cut-off mechanism (DCM) that adaptively allocates KV budgets across layers based on information density. Experiments on LongBench and ∞ Bench show consistent improvements over InfLLM, QLLM, and TokenSelect, achieving up to 10% accuracy gain with a 16× KV reduction. While slightly increasing latency, ActQKV offers a simple and effective way to enhance long-context reasoning and retrieval efficiency.

**Strengths:**

1. The paper introduces an activation-aware probe-query that leverages token-level activation bias to better capture semantic anchors within each window. The KV cut-off mechanism also adaptively allocates KV budgets across layers using information entropy, taking the sensitivity of different layers into consideration.
2. The proposed method is training-free and generalizable, easy to apply to multiple LLMs.
3. The method outperforms InfLLM, QLLM, and TokenSelect on existing long-context accuracy benchmarks under only 2K KV budget. The paper also includes detailed ablation and visualization showing how activation-aware selection better preserves models' performance.

**Weaknesses:**

1. The activation-bias computation and dynamic entropy-based cut-off introduce significant latency (1.6~1.9× slower than InfLLM), limiting real-time applicability.
2. The paper does not include comparisons with more recent retrieval-augmented or memory-optimized models (e.g., Quest, H2O).
3. While the paper introduces an “activation-aware probe-query,” the overall framework (importance-based KV retrieval) is conceptually close to existing attention sparsification and adaptive KV compression works (e.g., Ada-KV, PyramidKV, etc.).

**Questions:**

1. Is there a way to amortize or cache the activation bias computation across windows to reduce latency?
2. How is the overall latency (Table 10) distributed between different parts of ActQKV -- such as probe construction and dynamic KV recall?

---

### Official Review · Reviewer_GGF3 · 2025-10-31

**Soundness:** 3
**Presentation:** 2
**Contribution:** 2
**Rating:** 4
**Confidence:** 4

**Summary:**

This paper proposes ActQKV, a training-free KV-cache retrieval method for long-context LLMs. It builds an activation-aware probe-Query to identify salient tokens and uses per-layer entropy to adaptively allocate KV budgets during decoding. On LongBench and infinite-Bench, ActQKV achieves up to 16x KV reduction and improves accuracy.

**Strengths:**

1. Weighting Query vectors based on activation deviation is an interesting, training-free mechanism to surface semantically salient tokens within the current sliding window.
2. Adaptive KV allocation across layers, provide a more principled alternative to fixed top-k selection.
3. The inspection of cosine similarity distributions and perplexity behavior gives helpful insights into how retrieval behavior changes.

**Weaknesses:**

1. Computational overhead analysis in decoding, although the paper claims negligible overhead, no throughput or latency results are reported. This is critical in production inference.
2. Adaptive KV cache selection per layer leads to non-uniform attention shapes, which can: degrade kernel efficiency, complicate padding/masking, and reduce batching utilization. The paper does not evaluate real-world inference throughput under batched conditions.
3. When KV caching interacts with CPU/DRAM paging, retrieval cost would dominate, the scheduler should be discussed.

**Questions:**

1. Does activation bias correlate with attention distributions? Is there redundancy? Most training-free long context inference methods are based on attention sparsity.
2. How much average KV budget is actually allocated across layers during decoding? Whether the selected k is related the the depth of layer? Such as, early layers prefer larger k?
3. Can the authors report batch inference efficiency under typical serving batch sizes? Are attention kernels padded to the max? What is the GPU efficiency impact?
4. Does probe-Query computed in head-level or layer-level? For head-level, whether all the heads show the ability to detect important KV cache.

---

### Official Review · Reviewer_z8d7 · 2025-11-01

**Soundness:** 3
**Presentation:** 3
**Contribution:** 3
**Rating:** 4
**Confidence:** 4

**Summary:**

The idea of detecting “anchor” tokens via activation bias is interesting and addresses an important gap: many retrieval-methods treat all tokens uniformly.
Focusing the Probe-Query on tokens that deviate from the window mean gives a sharper context representation.

However, the paper doesn’t fully explore how retrieval behaves when the output is very long rather than short. Their experiments use long input / short output tasks, so applicability to long generation needs more validation.
Also, their notion of “sliding window” is loosely framed: in practice the windows appear to be blocks rather than fine-grained sliding strides, which may reduce continuity between windows and affect anchor detection near boundaries.

The proposed method relies on similarity matching between probe and key vectors, but don’t deeply discuss vector normalization vs dot-product subtleties, since Q×K in attention isn’t exactly cosine similarity, assumptions here may affect retrieval quality in practice.

In summary, the paper provides an easy to follow step in long context inference, especially in the matching stage of KV cache retrieval. However, it would be more convincing if the method were evaluated with reasoning models on long reasoning tasks.

**Strengths:**

1. Focuses on finding truly important tokens in long context, which gives a sharper and more useful probe for retrieval.
2. Training-free and easy to plug into existing long-context systems.
3. Shows notable accuracy gains under tight KV budgets.
4. Simple mechanism (activation bias) yet practical and effective.
5. Dynamic layer-wise KV budgeting is thoughtful and avoids wasting cache where it matters less.

**Weaknesses:**

1. Evaluations focus on long-input/short-output cases, so behavior in long-generation settings is unclear.
2. Windowing feels more like block windows rather than true sliding, which may lose continuity near boundaries.
3. Assumes dot-product similarity behaves like cosine without deeply addressing normalization or magnitude effects.
4. No latency/throughput reporting, so real deployment efficiency remains uncertain.

**Questions:**

1. Do you truly observe anchor tokens staying stable when outputs are long, or does the signal weaken in long-generation cases?
2. Have you compared token-level activation signals versus chunk-level aggregation? In real systems chunk-level often aligns better with KV grouping.
3. Since Q·K is not cosine, did you explicitly normalize vectors for retrieval, or rely on raw dot-product similarity?
4. Does activation bias sometimes highlight stylistic or formatting spikes rather than semantic anchors?
5. How does the method behave during very long decoding sequences where the active context evolves continuously?
6. What is the latency and memory overhead for computing activation bias across windows at scale?
7. Could head-specific signals improve anchor selection compared to layer-averaged bias?

---

### Official Review · Reviewer_HQVx · 2025-11-03

**Soundness:** 2
**Presentation:** 3
**Contribution:** 2
**Rating:** 2
**Confidence:** 4

**Summary:**

This paper proposes ActQKV, a training-free method to improve KV cache retrieval for long-context LLMs. It claims two main contributions: (1) An 'Activation-aware Probe-Query' (APQ) that weights query vectors based on an 'Activation Bias' to (in theory) better represent the context window, and (2) A 'Dynamic KV Cut-off Mechanism' (DCM) that allocates KV budgets per layer based on information density during decoding. Experiments on Long-Bench and $\infty$-Bench are presented to support these claims.

**Strengths:**

* The paper addresses an important and practical problem: KV cache efficiency for long-context inference.
* The authors are transparent about the method's latency overhead, which is reported in the appendix (Appendix D.5).
* The writing is clear and the proposed mechanism is easy to follow.

**Weaknesses:**

1.  **Missing Evaluation on Long-Decoding Scenarios:** The paper's entire evaluation is on benchmarks like Long-Bench, which are primarily "long-prompt, short-answer" tasks (e.g., document QA). This setup completely misses the main bottleneck of current LLMs: **long-decoding scenarios**. These are tasks like long-form chain-of-thought reasoning, where the prompt might be short, but the *generated content* becomes extremely long. As the model generates thousands of tokens, its *own* KV cache (from its *own* output) becomes the bottleneck. This method is never tested on this critical "short-prompt, long-generation" workload, making its practical utility highly questionable.
2.  **Lack of Reasoning Task Evaluation:** Related to the first point, the method's impact on complex, multi-step reasoning is completely unknown. The authors should have added experiments on strong reasoning models (like DeepSeek-R1 or Qwen3) on reasoning-specific tasks to see if this retrieval method actually helps or hurts the quality of a long, generated chain of thoughts.
3.  **Incremental Contribution:** The core ideas are not very original. Using an importance-weighted sum to create a query vector, or dynamically adjusting a budget based on layer-wise statistics, are both fairly straightforward extensions of existing work. The performance gains do not seem to justify the added complexity.
4.  **Significant Latency Overhead:** The method, by the authors' own admission in Table 10, **slows things down considerably** (1.6-1.9x slower than the InfLLM baseline). This is a massive practical disadvantage. A method that nearly doubles inference time is a non-starter for most real-world applications, and the paper does not provide a strong enough justification for this trade-off.

**Questions:**

1.  Why did you choose to only evaluate on long-prompt, short-answer tasks? The main bottleneck for many LLM applications is long-form *generation* (long decoding). Can you provide any data on how your method performs in a long chain-of-thought reasoning task, where the generated content far exceeds the prompt length?
2.  Can you add experiments on reasoning-focused models, such as DeepSeek-R1, to demonstrate that your retrieval mechanism does not harm (and ideally, helps) the complex, step-by-step generation required for these tasks?
3.  Given the 1.6-1.9x latency overhead, how do you justify the practical applicability of this method? This severe slowdown seems to hinder, not help, its application.

---

### Note · Authors · 2026-01-05

I have read and agree with the venue's withdrawal policy on behalf of myself and my co-authors.